# Industrial Low-Clinker Precast Elements Using Recycled Aggregates

**Carlos Thomas** [1,*], **Ana I. Cimentada** [1], **Blas Cantero** [2], **Isabel F. Sáez del Bosque** [2] **and Juan A. Polanco** [1]

[1] LADICIM (Laboratory of Materials Science and Engineering), University of Cantabria E.T.S. de Ingenieros de Caminos, Canales y Puertos, Av. Los Castros 44, 39005 Santander, Spain; anaisabel.cimentada@unican.es (A.I.C.); polancoa@unican.es (J.A.P.)

[2] School of Civil Engineering, University of Extremadura, Institute for Sustainable Regional Development (INTERRA), Avda. de la Universidad, s/n, 10003 Cáceres, Spain; bcanteroch@unex.es (B.C.); cmedinam@unex.es (I.F.S.d.B.)

[*] Correspondence: thomasc@unican.es

**Abstract:** Increasing amounts of sustainable concretes are being used as society becomes more aware of the environment. This paper attempts to evaluate the properties of precast concrete elements formed with recycled coarse aggregate and low clinker content cement using recycled additions. To this end, six different mix proportions were characterized: a reference concrete; 2 concretes with 25%wt. and 50%wt. substitution of coarse aggregate made using mixed construction and demolition wastes; and others with recycled cement with low clinker content. The compressive strength, the elastic modulus, and the durability indicator decrease with the proportions of recycled aggregate replacing aggregate, and it is accentuated with the incorporation of recycled cement. However, all the precast elements tested show good performance with slight reduction in the mechanical properties. To confirm the appropriate behaviour of New Jersey precast barriers, a test that simulated the impact of a vehicle was carried out.

**Keywords:** recycled concrete; low clinker cement; precast; mechanical properties; physical properties; New Jersey barriers

## 1. Introduction

Construction and demolition waste (CDW) is non-hazardous, inert waste generated in any construction, rehabilitation or demolition work. The industrial and construction sectors generate practically the same amount of non-hazardous waste (industry 37,417 kt[‡] and construction 35,869 kt[‡]) in Spain [1]. The European Commission estimates that the volume of CDW comprises one third of all waste generated in the European Union, which constitutes the largest waste stream [2]. Recycling this CDW would lead to more sustainable growth, replacing a linear economy based on use of materials with a more circular economy. This is important, as aggregates are the second-most-used raw material by humans, behind only water [3]. There is European legislation to encourage recycling CDW [4] and many countries have specific norms for the use of recycled aggregates (RA) for concrete [5–8]. In addition, the use of RA could lead to cheaper concrete [9].

Several studies have corroborated that the inclusion of RA produces concrete with a lower density and increased heterogeneity [10–12]. RA normally has a higher porosity than natural aggregate (NA) [13]. In a fresh state, Silva et al. [11] concluded that recycled aggregate concrete (RAC) is less workable and, to achieve a workability equivalent to that of NA, RA could be pre-saturated, or water added during mixing to compensate [14]. However, the incorporation of completely saturated

aggregates might cause an excessive water supply [15,16]. Once the RAC hardens, these aggregates make the concrete more susceptible to detrimental environmental effects, resulting in a lower durability [17,18], which should be taken into consideration. Consequently, Annex 15 of the Spanish Instruction for Structural Concrete EHE-08 [19] and other studies [14,20] propose solutions, such as increasing the cement content, reducing the water/cement ratio, or increasing the coating thickness in the case of reinforced concrete.

Generally, it is known that the incorporation of RA into concrete reduces its mechanical properties [21,22], due to the presence of contaminants such as plastics, glass, adhered mortar, etc., ref. [23] and the type of source material (crushed concrete, ceramic or mixed) of the RA [24–26]. The elastic modulus of RAC is lower than that of conventional concrete [15], reaching 45% less for 100% replacement [25]. The results obtained in the characterization of RAC with intermediate replacements present greater variation of results [20]. Other authors have demonstrated the viability of other types of recycled aggregates from waste, such as steel slag [27]. Moreover, the RA affects the fatigue behavior of the concrete [28–32], showing a greater loss of properties than with the static properties. Further research has evaluated the recycling of concrete which incorporates RA [33,34].

With regard to precast concrete elements, it should be noted that, according to the ANDECE (National Association of the Prefabricated Concrete Industry, based in Spain), although the initial cost of elements is higher, the final cost is lower [35]. Other studies such as López-Mesa et al. [36] indicate an almost 18% higher cost of precast slabs versus in situ slabs; although the former have a lower environmental impact and the quality may be higher. Normally, precast elements have a quality seal guaranteeing their properties. Due to a manufacturing process with complete exhaustive control, precast slabs can be: tailored with special properties more easily as they are not manufactured on site; designed with flexibility difficult to achieve in-situ; and incorporate RA in their fabrication. In the case of precast elements using RA, a lower density and strength is observed [37]. Poon et al. [37] investigated the factors that affect the properties of precast concrete blocks with RA, concluding that the compressive strength increases with the reduction in the aggregate/cement ratio (A/C), and that the water absorption of concrete blocks is significantly related to the absorption capacity of the aggregate. Katz [21] investigated the use of precast elements at different ages to produce RA for new precast elements, concluding that the mechanical properties (strength, modulus of elasticity, etc.) when using this type of aggregate in concrete, resemble those when using lightweight aggregates, such as those manufactured using fly ash.

This paper presents the effect on physical and mechanical properties of six types of mixes with different degrees of substitution. The physical properties and durability of these concretes will be analyzed first, then the mechanical properties will be assessed. Finally, the behavior of precast elements will be addressed.

## 2. Materials and Methodology

The natural siliceous aggregate used in this study is present in three different sizes: 6/0 mm (NS), 12/6 mm (NG-M), and 22/12 mm (NG-C). Mixed recycled aggregates (MRA) were used by substituting NG-M for MRA-M and NG-C for MRA-C. These MRA were obtained from CDW and were principally made up of concrete and mortars ($\approx$ 45%), unbound aggregate, and natural stone ($\approx$ 45%). Figure 1 shows the different size grading for each aggregate.

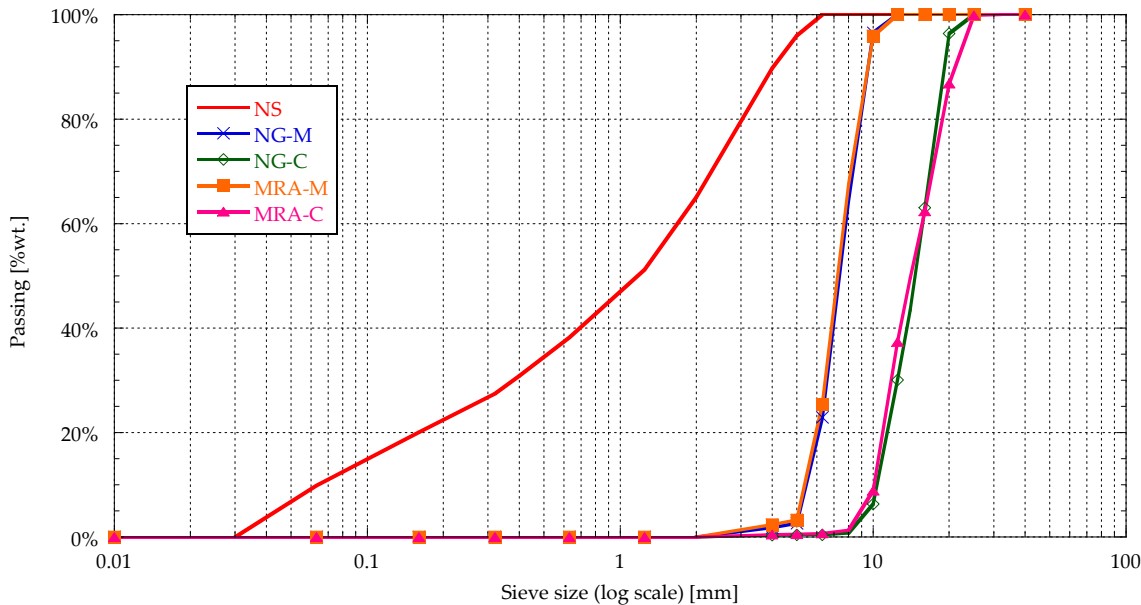

**Figure 1.** Grading of the aggregates.

Table 1 displays physical and mechanical properties: where *SSS* is the saturated dry surface density according to EN 1097-6 [38]; *A* is the water absorption by weight according to EN 1097-6 [38]; *LA* is the Los Angeles index according to EN 1097-2 [39]; and *FI* is the flakiness index according to EN 933-3 [40].

**Table 1.** Physical and mechanical properties of the aggregates.

| Property | Aggregates | | | | |
|---|---|---|---|---|---|
| | **NS** | **NG-M** | **NG-C** | **MRA-M** | **MRA-C** |
| *SSS* [g/cm$^3$] | 2.76 | 2.74 | 2.74 | 2.42 | 2.45 |
| *A* [%] | 1.18 | 0.88 | 0.78 | 6.28 | 5.27 |
| *LA* [%] | - | 16 | 18 | 32 | 36 |
| *FI* [%] | - | 21 | 25 | 10 | 10 |

The conventional cement (OPC) was CEM I 42.5 R, and the low clinker content cement (RC) was constituted of 75% CEM I 42.5 R and 25% ceramic waste from CDW. The tests performed with the cement revealed a compressive strength 20% higher in the case of OPC.

Mixing the aggregates in different proportions with the two existing types of cement produced six concrete mixtures, as shown in Table 2. HP signifies a combination of natural aggregates and conventional cement. HPR is a mixture of natural aggregates and low clinker content cement. HR25 and HR50 were fabricated with conventional cement and substitutions of NA by 25%wt. and 50%wt. proportions of RA, respectively. Finally, HRR25 and HRR50 were obtained by amalgamating low clinker content cement with natural aggregates, substituted by 25%wt. and 50%wt. of recycled aggregates accordingly.

**Table 2.** Concrete mix proportions (by $m^3$).

| Concrete: | HP | HPR | HR25 | HR50 | HRR25 | HRR50 |
|---|---|---|---|---|---|---|
| NS (6/0 mm) [kg]: | 732 | 732 | 719 | 705 | 719 | 705 |
| NG-M (12/6 mm) [kg]: | 382 | 382 | 284 | 184 | 284 | 184 |
| NG-C (22/12 mm) [kg]: | 766 | 766 | 568 | 369 | 568 | 369 |
| MRA-M (12/6 mm) [kg]: | - | - | 89 | 178 | 89 | 178 |
| MRA-C (22/12 mm) [kg]: | - | - | 178 | 356 | 178 | 356 |
| Cement [kg]: | 400 | - | 400 | 400 | - | - |
| Low clinker content cement [kg]: | - | 400 | - | - | 400 | 400 |
| Water [kg]: | 193 | 193 | 202 | 211 | 202 | 211 |
| Superplasticizer [kg]: | 6.2 | 6.2 | 6.2 | 6.2 | 6.2 | 6.2 |
| Water/cement ratio | 0.48 | 0.48 | 0.50 | 0.53 | 0.50 | 0.53 |

## 2.1. Physical and Mechanical Properties

Densities were obtained according to EN-12390-7 [41]. Sub-specimens (10Ø × 10 cm) obtained by cutting 10Ø × 20 cm cylindrical specimens were used. The porosity coefficient is the result of comparing the absorbed water and specimen volume, while the absorption coefficient is the result of comparing the absorbed water and specimen weight. Compressive strength was determined using 10Ø × 20 cm cylindrical specimens according to EN-12390-3 [42], with an application strength rate of 0.5 MPa/s. Elastic modulus was determined with 10Ø × 20 cm cylindrical specimens according to EN-12390-13 [43], at a strength rate of 0.5 MPa/s.

## 2.2. Durability

A water penetration test was performed according to EN-12390-8 [44]. Sub-specimens (10Ø × 10 cm) obtained by cutting 10Ø × 20 cm cylindrical specimens were used. The samples were subjected to a pressure of 5 bar for 72 h. After 72 h water penetration under pressure, it was necessary to analyze how deep the water reached. To be able to observe the interior of the sample, it had to be opened. During this research, the Brazilian method (or indirect tensile strength method) was used to open the sample and analyze its interior. In general, when a cylindrical specimen is subjected to tension along its generatrix, it breaks into two halves, which allows the interior to be analyzed. Once the specimen had been opened, it was possible to measure the penetration depth of the water into the porous concrete. This technique also provided another interesting result: the indirect tensile strength of the concrete. For the determination of oxygen permeability, UNE-83981 [45] was taken as a reference. The 10Ø × 20 cm cylindrical specimens were cut to discard the upper and lower face obtaining a new sample of 10Ø × 10 cm. Silicone was impregnated perimetrically in the samples so that the oxygen could only pass longitudinally. A regulated oxygen pressure was applied on the upper face. Digital flow meters registered the oxygen escaping from the lower face.

## 2.3. Precast Element Preparation

Two different types of precast elements were manufactured: unreinforced concrete ditches and steel-reinforced New Jersey barriers. Both were manufactured with an industrial concrete mixer, poured in metallic molds and vibrated by hand (Figure 2). In the case of reinforced concrete, reinforcements were set into the mold before the pouring of concrete. In both cases, precast elements were unmolded and cured at ambient temperature.

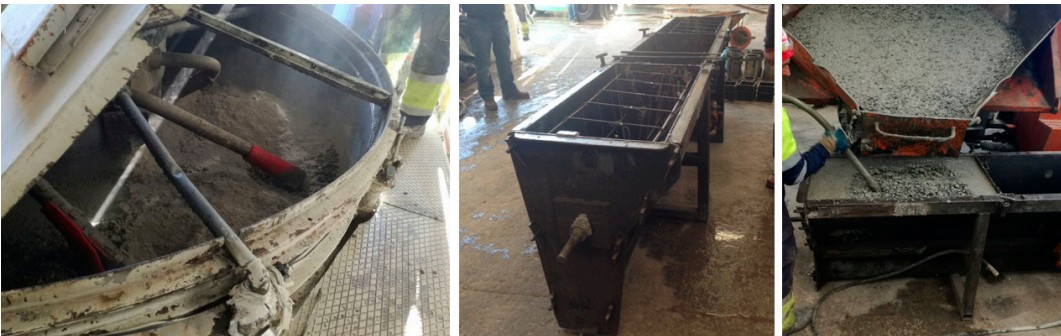

**Figure 2.** Precast element manufacturing sequence.

### 2.4. Precast Element Mechanical Characterization

Concrete ditches have approximate measurements of $50 \times 50 \times 15$ cm. In order to characterize concrete ditches, the tests were carried out by bending. The horizontality of the set was verified, and force was applied by a roller ($10\varnothing \times 22$ cm) in the central section with a displacement rate of 0.1 mm/s (Figure 3).

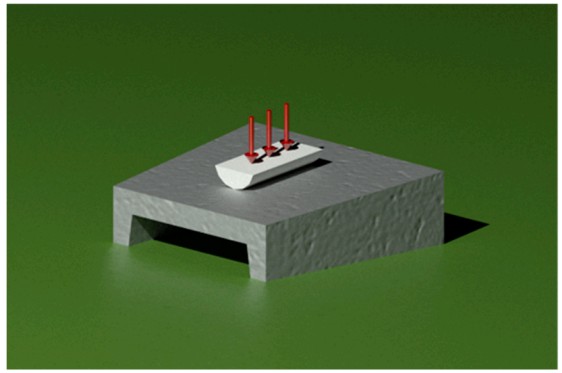
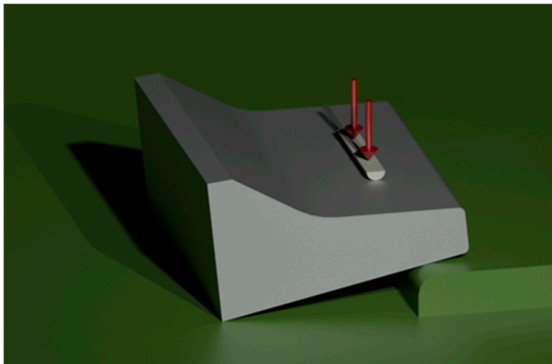

**Figure 3.** Precast element characterization (concrete ditches left, New Jersey barriers right).

New Jersey barriers have a section with approximate measurements of $47 \times 80$ cm and a length of 100 cm. In order to characterize New Jersey barriers, a small crane was used to support the precast element on steel beams. These steel beams were placed at one end to correct the inclination of the face on which the test was to be performed, achieving horizontality on that face (Figure 3). The test consisted in applying a stress with a roller ($3\varnothing \times 40$ cm). The time of the test was very short (0.1–0.2 s) to simulate an impact. The strength and displacement data of the actuator were recorded during the test.

## 3. Results and Discussion

### 3.1. Physical Properties

Figure 4 shows the relative and saturated densities of the concretes. As demonstrated, the density decreases as the percentage of NA replaced by RA increases. This is due to the lower density of RA. It also becomes clear that the use of this RC does not affect density significantly.

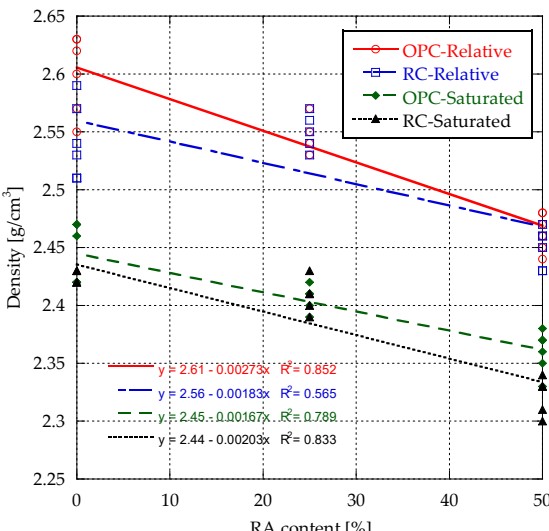

**Figure 4.** Density vs. RA content.

Figure 5a shows porosity, and Figure 5b shows the absorption coefficient vs. substitution of NA by RA. A decrease in both properties is found in the concretes containing OPC as the percentage of replacement of aggregate increases. However, in the case of concrete made with RC, both properties increase as the percentage of RA increases. This may be because this type of cement interacts more with RAs of different nature, making it difficult to fill all the gaps amongst aggregates. Alternatively, it may be because the RA is able to absorb more water during kneading, causing a small deficit in this type of cement, which is very susceptible to variations in the water dosage. It is possible that there may be another reason that has not been identified.

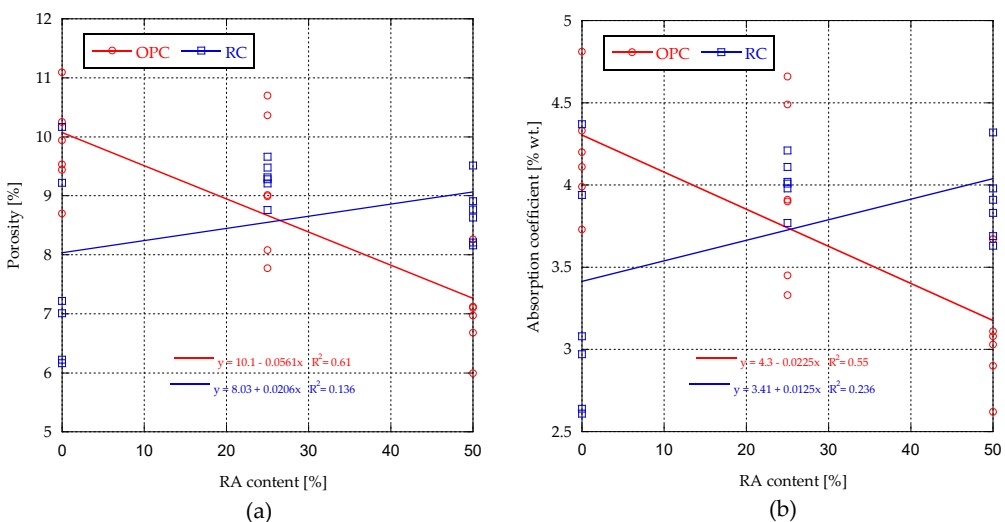

**Figure 5.** Porosity (**a**) and absorption coefficient (**b**) vs. RA content.

### 3.2. Compressive Strength and Modulus of Elasticity

Figure 6a shows the compressive strength-strain curves for each concrete at 160 days. Several studies [25,46,47] show that the concrete's compressive strength decreases with the degree of substitution of RA for NA, but in strain terms, concretes show similar values around 2500 μm/m for the failure. The exception is the HRR50 mix, which exceeds the values of the rest by almost 1000 μm/m. Figure 6b shows the same mixtures but at an age of 365 days. The decrease in strength may also be due to

the randomness of the type of RA and its distribution into the mortar matrix, which causes greater uncertainty than conventional mixtures.

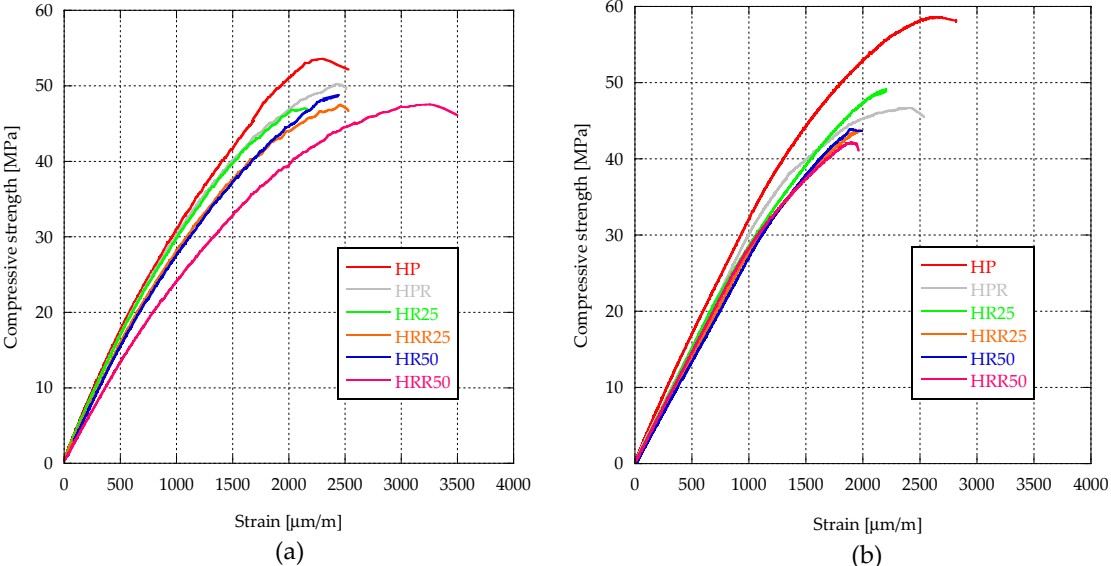

**Figure 6.** Compressive strength-strain at 160 (**a**) and 365 (**b**) days.

Table 3 shows the different values of compressive strength obtained at different ages.

**Table 3.** Compressive strength at different ages.

| Concrete: | Compressive Strength [MPa] | | | |
|---|---|---|---|---|
| | 28 days | 160 days | 365 days | $\Delta_{36-28}$ [%] |
| HP | 51.2 | 53.5 | 56.8 | +10.9 |
| HPR | 46.1 | 50.2 | 46.6 | +1.1 |
| HR25 | 51.7 | 47.0 | 45.1 | −12.8 |
| HR50 | 51.2 | 48.8 | 42.1 | −17.7 |
| HRR25 | 45.0 | 47.5 | 43.7 | −2.9 |
| HRR50 | 41.2 | 47.5 | 42.4 | +2.9 |

Table 4 displays the modulus of elasticity, and shows that when using RC, the decrease in the elastic modulus is around 4%. The substitution of 25% by RA implies a decrease in elastic modulus of 5.6%, while the substitution of OPC in this case does not seem to have an influence. In the case of replacing 50% of aggregate by RA, the influence of the substitution of OPC by RC is meaningful, decreasing the elastic modulus by 15%. As for the loss of elastic modulus over time, a greater influence of the cement is observed than the type of aggregate, with a limit that tends to an asymptotic value of around 27 GPa.

**Table 4.** Modulus of elasticity.

| Concrete: | Substitution [%] | Modulus of Elasticity [GPa] | Modulus of Elasticity at 365 days [GPa] | % of the Initial Elastic Modulus |
|---|---|---|---|---|
| HP | 0 | 35.5 | 31.7 | 89.3 |
| HPR | 0 | 34.1 | 29.5 | 86.5 |
| HR25 | 25 | 33.9 | 30.8 | 90.8 |
| HR50 | 50 | 31.9 | 29.3 | 91.8 |
| HRR25 | 25 | 34.2 | 27.9 | 81.6 |
| HRR50 | 50 | 27.8 | 27.4 | 98.6 |

Some organizations such as EHE-08, ACI, and Eurocode present their expressions to predict elastic modulus at 28 days from the compressive strength. In Expressions (1)–(3): *E* is elastic modulus at 28 days [GPa] and $f_{28}$ is the compressive strength at 28 days [MPa].

EHE-08 [48]

$$E = 8.5 \sqrt[3]{f_{28}} \tag{1}$$

ACI [49]

$$E = 4.7 \sqrt{f_{28}} \tag{2}$$

Eurocode 2 [50]

$$E = 22(f_{28}/10)^{0.3} \tag{3}$$

These expressions can be used to obtain the predictions and comparisons, with the experimental results shown in Table 5. The ACI method fits quite well in most cases but predicts higher values when the percentage of substitution is 50%. The EHE-08 method is safer, although when the substitution is 50% and the OPC is replaced by RC, higher values are produced due to the heterogeneity of the RA affecting the compressive strength. These types of expressions only satisfactorily fit ordinary concrete models.

**Table 5.** Elastic modulus obtained with different expressions.

| Concrete: | Elastic Modulus [GPa] | | | | |
|---|---|---|---|---|---|
| | Experimental | EHE-08 | ACI | Eurocode 2 | $\Delta_{\text{experimental−EHE-08}}$ [%] |
| **HP** | 35.5 | 31.6 | 33.6 | 35.9 | 12.5 |
| **HPR** | 34.1 | 30.5 | 31.9 | 34.8 | 11.9 |
| **HR25** | 33.9 | 31.7 | 33.8 | 36.0 | 7.1 |
| **HR50** | 31.9 | 31.6 | 33.6 | 35.9 | 1.1 |
| **HRR25** | 34.2 | 30.2 | 31.5 | 34.5 | 13.1 |
| **HRR50** | 27.8 | 29.4 | 30.2 | 33.6 | -5.3 |

Figure 7 shows that from approximately 48 MPa, concrete with RA achieved the same compressive strength as concrete with OPC. RA concrete increases its elastic modulus significantly. This might be due to the addition of a new variable, such as RA compared with OPC, which is much more standardized throughout its production process.

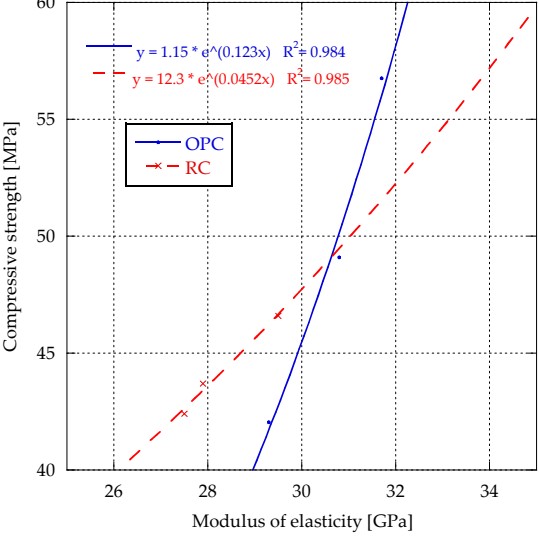

**Figure 7.** Compressive strength vs. modulus of elasticity.

### 3.3. Oxygen and Water Permeability

Figure 8a shows the oxygen permeability and Figure 8b shows the maximum penetration of water vs. percentage of substitution, respectively.

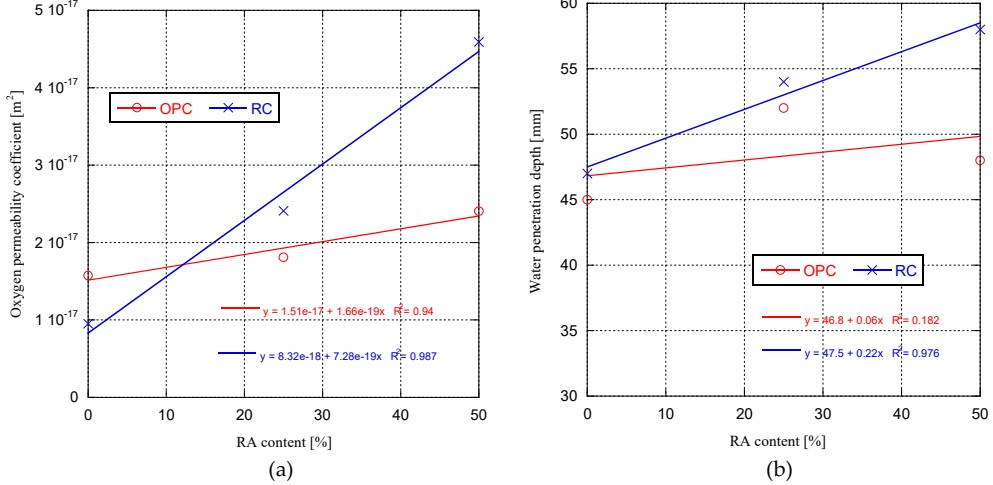

(a)    (b)

**Figure 8.** Oxygen permeability coefficient (**a**) and water penetration depth (**b**).

The oxygen permeability coefficient increases with the substitution of the NA by RA. This behavior has been reported in some studies, such as Ismail et al. [51], Medina et al. [52], and Thomas et al. [14]. This increase is higher in concrete with RC than OPC; the type of cement being used is an important factor.

The penetration of water increases with the increase in RA substitution. With these results, only HP and HPR comply with the standard EHE-08 [48] for structural concrete in the case of IIIa, IIIb, IV, etc. environment exposition, which requires an average penetration depth of 30mm, and maximum penetration depth of 50 mm. Penetration of water is related to typology and distribution of the RA, and its impurities with high absorption coefficients.

Figure 9 shows cross-sections of concrete where different colors can be seen. These are caused by the RC in HPR and HRR50 mixtures, and some kind of RA and impurities (such as wood or fired clay) in HRR50 mix.

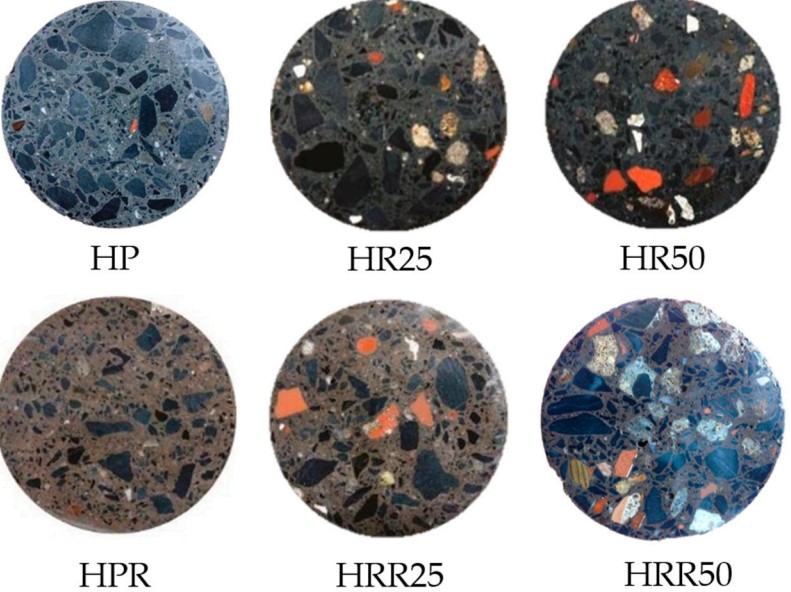

**Figure 9.** Concrete specimen sections.

### 3.4. Testing Precast Elements

Figure 10a shows the results of flexural tests on concrete ditches. It can be observed that the concrete composed of RC and RA (HRR50) behaves similarly to HP concrete, which is consistent with the results of splitting tensile strength shown in Table 6. Figure 10b shows the results of the impact test on reinforced precast New Jersey barriers, in which the force applied by the test machine and the position of the actuator are recorded. As expected, the concrete with OPC and NA displayed superior mechanical behavior than concrete with RC and RA. HRR50 could resist only 60% of the force, and 66% of the displacement that HP resisted.

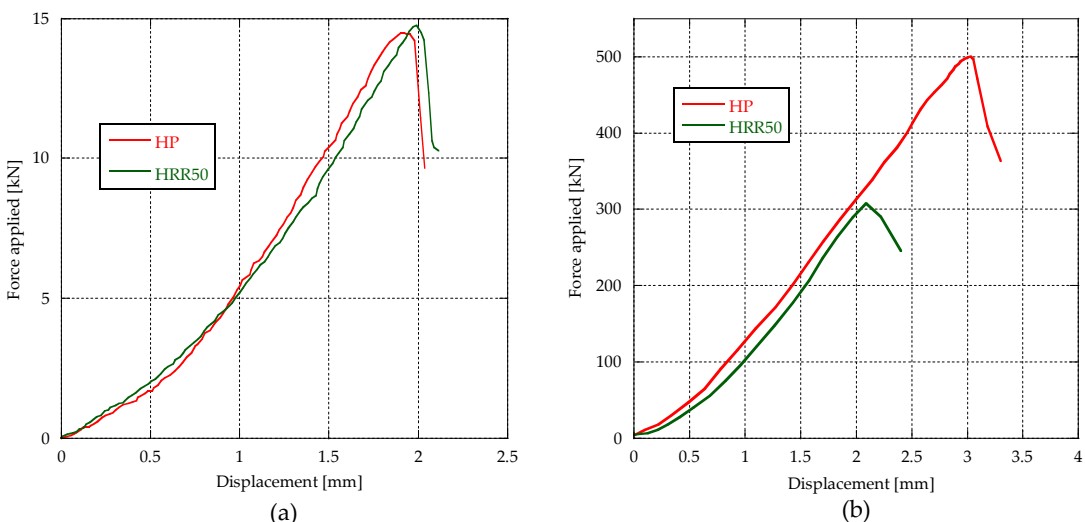

(a)                                                    (b)

**Figure 10.** Mechanical characterization of precast elements: Bending test on ditches (**a**), impact test on barriers (**b**).

**Table 6.** Splitting tensile strength.

| Splitting Tensile Strength [MPa] | | | | | |
|---|---|---|---|---|---|
| **HP** | **HPR** | **HR25** | **HR50** | **HRR25** | **HRR50** |
| **3.36** | 3.51 | 3.48 | - | 3.30 | 3.58 |

Figure 11 shows the results of the test performed with both types of precast elements. Different sections of cracks in OPC and RC concrete ditches, and the fissure produced in a New Jersey barrier are visual results of the tests.

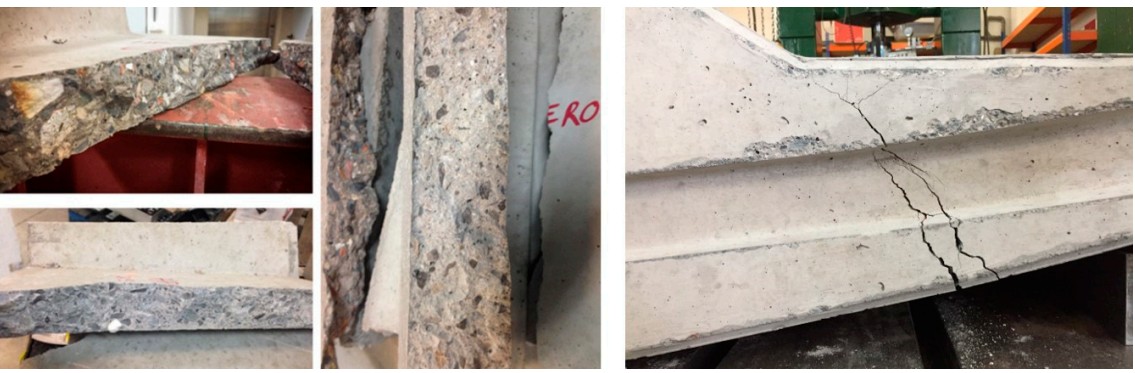

**Figure 11.** Precast test and cracking.

Equation (4) indicates whether a New Jersey barrier could withstand the perpendicular impact of a vehicle. Velocity and mass are variables, and it would be necessary to incorporate a restitution coefficient in order to avoid the elastic impact.

This coefficient relates the velocity before impact with the velocity after collision, considering the barrier is without velocity before and after impact.

$$C_R = -\frac{V_{1f} - V_{2f}}{V_{1i} - V_{2i}}; when\ V_{2f}, V_{2i} = 0 \rightarrow C_R = -\frac{V_f}{V_i} \tag{4}$$

García and Cabreiro [53] proposed a method for obtaining the coefficient of restitution based on experimental processes in "*Use of dynamic models in the investigation of road accidents*" (text in Spanish), for which they suggested two equations:

$$C_R = 0.45 \cdot e^{(-0.040278 \cdot v)}, For\ v\ <\ 54\ km/h \tag{5}$$

$$C_R = 0.45 \cdot e^{(-0.015278 \cdot v)}, For\ v\ \geq\ 54\ km/h \tag{6}$$

With Equations (4)–(6), considering the maximum force that a barrier resists, and the duration of the impact as 0.1 s, Equations (7) and (8) are obtained, shown in Figure 12.

$$m = \frac{0.1 \cdot F}{-\left(\frac{v_i}{3.6}\right) \cdot \left(0.45 \cdot e^{(-0.040278 \cdot v)} + 1\right)}, For\ v\ <\ 54\ km/h \tag{7}$$

$$m = \frac{0.1 \cdot F}{-\left(\frac{v_i}{3.6}\right) \cdot \left(0.12 \cdot e^{(-0.015278 \cdot v)} + 1\right)}, For\ v\ \geq\ 54\ km/h \tag{8}$$

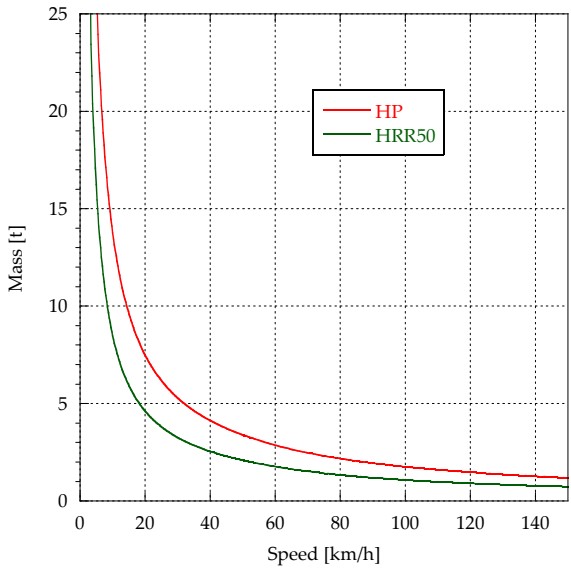

**Figure 12.** Simulated behavior of reinforced barriers.

These curves are conservative, as the barrier can withstand strains that absorb energy before cracking, and the parapet would not always be immobile (they are only anchored to the ground on viaducts).

## 4. Conclusions

Characterization tests on concrete specimens and precast elements have been carried out using low-clinker cements and recycled aggregates, obtaining the following conclusions. Firstly, the physical-mechanical properties of mixed recycled aggregates are suitable for the manufacture of

concrete and precast elements when the medium and coarse fraction is used. Secondly, the use of mixed recycled aggregates causes a loss of density and compressive strength slightly higher than that which occurs when using recycled concrete aggregates. Recycled concretes made from low-clinker cement are slightly more porous than concretes made with ordinary Portland cement. Finally, regarding the mechanical properties of recycled concrete, a loss of around 10% of the compressive strength is observed when using low-clinker cement. In addition, recycled concrete made with ordinary Portland cement evolves slightly more when over 1 year of curing has elapsed.

**Author Contributions:** Conceptualization, C.T., A.I.C., J.A.P.; methodology, C.T., A.I.C., J.A.P., I.F.S.d.B., B.C.; validation, C.T., I.F.S.d.B., B.C.; formal analysis, C.T.; investigation, C.T., A.I.C., J.A.P.; resources, C.T., J.A.P.; writing—original draft preparation, C.T., I.F.S.d.B., B.C.; writing—review and editing C.T. All authors have read and agreed to the published version of the manuscript.

**Funding:** This research was funded by SODERCAN, S.A. (SODERCAN/FEDER) and BIA2013-48876-C3-2-R awarded by the Ministry of Science and Innovation.

**Acknowledgments:** The authors would like to express our gratitude to Jaime de la Fuente and César Medina for their support and participation in part of the project.

**Conflicts of Interest:** Authors declare no conflict of interest.

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
