# Peer review of "Industrial Low-Clinker Precast Elements Using Recycled Aggregates"

_applsci, doi:10.3390/app10196655_

Round 1

Reviewer 1 Report

This study investigated a combination of recycled aggregates and recycled cement with low clinker content. The test design is well designed. It is interesting to include the strength of 365 d. A few comments:

  1. Line 55, AR is not defined before;
  2. Line 117, "After that, samples were submitted to indirect tensile stress to clearly see the maximum depth of water penetration and mark it." it is not easy to understand.
  3. line 263, some equations are not in a good shape, should be revised.

Author Response

Dear Reviewer,

We greatly appreciate the opportunity you give us to improve the paper with your valuable comments. Here you can find the detailed comments and performed changes.

Best regards,

The authors

Responses to comments:

Reviewer #1

This study investigated a combination of recycled aggregates and recycled cement with low clinker content. The test design is well designed. It is interesting to include the strength of 365 d.

Thank you very much for this comment. The compressive strength at 365 days has been included in the Table 3. Compressive strength at different ages.

A few comments:

  1. Line 55, AR is not defined before;

The reviewer is wright, “AR” is a mistake. We have used “AR” instead “RA”. The sentence has been corrected: Other researches have evaluated the recycling of concrete that incorporates RA

  1. Line 117, "After that, samples were submitted to indirect tensile stress to clearly see the maximum depth of water penetration and mark it." it is not easy to understand.

The reviewer is right. The sentence is confusing. We have replaced it with this other one for clarification. “After 72 hours of penetration of water under pressure, it is necessary to analyze the depth reached by the water. To be able to observe the interior of the sample, it has to be opened. The method used on this research to open the sample and analyze its interior has been the Brazilian method or indirect tensile strength method. In general, when a cylindrical specimen is subjected to tension along its generatrix, it breaks into two halves that allow the interior to be analyzed. When the specimen has been opened, it is possible to measure the penetration depth of the water into the porous concrete. This technique also provides another d interesting result: the indirect tensile strength of the concrete.”

  1. line 263, some equations are not in a good shape, should be revised.

Indeed, comma and period errors and an notation have been corrected. Equations 3.4 to 3.7 have been changed to (see manuscript):

For v < 54 km/h

(3.5)

For v ≥ 54 km/h

(3.6)

For v < 54 km/h

(3.7)

For v ≥ 54 km/h

(3.8)

Reviewer 2 Report

The topic is interesting and has a high quality.

Author Response

Responses to the reviewers’ comments to the manuscript

Industrial low-clinker precast elements using recycled aggregates

----

Dear Reviewer,

We greatly appreciate the opportunity you give us to improve the paper with your valuable comments. Here you can find the detailed comments and performed changes.

Best regards,

The authors

Responses to comments:

Reviewer #2

The topic is interesting and has a high quality.

Thank you very much for your comment.

Reviewer 3 Report

Dear authors:

Your paper is good, however your last conclusion is not supported in your paper. I do believe that you found greater dispersion but in your paper you did not show that. Therefore you have to remove this conclusion or alternatively (better) you have to show that in results, analyse and discuss that in order to get a conclusion fully supported.

Line 272, replace 'one' by 'on'.

Best regards

Author Response

Responses to the reviewers’ comments to the manuscript

Industrial low-clinker precast elements using recycled aggregates

----

Dear Reviewer,

We greatly appreciate the opportunity you give us to improve the paper with your valuable comments. Here you can find the detailed comments and performed changes.

Best regards,

The authors

Responses to comments:

Reviewer #3

Dear authors:

Your paper is good, however your last conclusion is not supported in your paper. I do believe that you found greater dispersion but in your paper you did not show that. Therefore you have to remove this conclusion or alternatively (better) you have to show that in results, analyse and discuss that in order to get a conclusion fully supported.

Thank you very much for your comment. We agree with you and the conclusion not supported by the results has been removed.

Line 272, replace 'one' by 'on'.

The reviewer is right. It was a mistake. "One" has been replaced by "On", resulting the sentence: “Characterization tests on concrete specimens and precast elements ...”

Best regards
